# Multi-insecticide resistant malaria vectors in the field remain susceptible to malathion, despite the presence of *Ace1* point mutations

Nadja C. Wipf[1,2]*, Wandrille Duchemin[3], France-Paraudie A. Kouadio[1,2,4,5], Behi K. Fodjo[4,5], Christabelle G. Sadia[4,5], Chouaïbou S. Mouhamadou[4,6], Laura Vavassori[1,2,4], Pascal Mäser[1,2ʘ], Konstantinos Mavridis[7ʘ], John Vontas[7,8ʘ], Pie Müller[1,2]*

1 Swiss Tropical and Public Health Institute, Basel, Switzerland, 2 University of Basel, Basel, Switzerland, 3 Center for Scientific Computing (sciCORE), University of Basel, Basel, Switzerland, 4 Centre Suisse de Recherches Scientifiques en Côte d'Ivoire, Abidjan, Côte d'Ivoire, 5 Université Nangui Abrogoua, Abidjan, Côte d'Ivoire, 6 Department of Entomology and Plant Pathology, North Carolina State University, Raleigh, North Carolina, United States of America, 7 Institute of Molecular Biology and Biotechnology, Foundation for Research and Technology-Hellas, Heraklion, Greece, 8 Pesticide Science Laboratory, Department of Crop Science, Agricultural University of Athens, Athens, Greece

ʘ These authors contributed equally to this work.
* nadja.wipf@swisstph.ch (NCW); pie.mueller@swisstph.ch (PM)

**Data Availability Statement:** The raw RNA sequencing data and related metadata are available from the public repository NCBI Sequence Read

## Abstract

Insecticide resistance in *Anopheles* mosquitoes is seriously threatening the success of insecticide-based malaria vector control. Surveillance of insecticide resistance in mosquito populations and identifying the underlying mechanisms enables optimisation of vector control strategies. Here, we investigated the molecular mechanisms of insecticide resistance in three *Anopheles coluzzii* field populations from southern Côte d'Ivoire, including Agboville, Dabou and Tiassalé. All three populations were resistant to bendiocarb, deltamethrin and DDT, but not or only very weakly resistant to malathion. The absence of malathion resistance is an unexpected result because we found the acetylcholinesterase mutation *Ace1*-G280S at high frequencies, which would typically confer cross-resistance to carbamates and organophosphates, including malathion. Notably, Tiassalé was the most susceptible population to malathion while being the most resistant one to the pyrethroid deltamethrin. The resistance ratio to deltamethrin between Tiassalé and the laboratory reference colony was 1,800 fold. By sequencing the transcriptome of individual mosquitoes, we found numerous cytochrome P450-dependent monooxygenases – including CYP6M2, CYP6P2, CYP6P3, CYP6P4 and CYP6P5 – overexpressed in all three field populations. This could be an indication for negative cross-resistance caused by overexpression of pyrethroid-detoxifying cytochrome P450s that may activate pro-insecticides, thereby increasing malathion susceptibility. In addition to the P450s, we found several overexpressed carboxylesterases, glutathione *S*-transferases and other candidates putatively involved in insecticide resistance.

Archive (SRA) under the BioProject Accession No. PRJNA772169. The R scripts used to generate analyses, figures and tables are available from the GitHub repository https://github.com/NadjaWipf/NCR_CI.

**Funding:** This work was funded by the European Research Council (ERC) under the European Union's Horizon 2020 research and innovation programme: ICT-39-2015 - International partnership building in low and middle income countries (Grant Agreement No. 688207 – DMC-MALVEC) to KM, JV and PMR; and the Novartis Foundation for Medical-Biological Research (No. 19B134) to NW and PMR. The Voluntary Academic Society Basel / Freiwillige Akademische Gesellschaft Basel (FAG) and the Rudolf Geigy Foundation / R. Geigy-Stiftung supported NW. We are thankful for the support by the Research Infrastructures for the control of vector-borne diseases that has received funding from the European Union's Horizon 2020 research and innovation programme (Grant Agreement No. 731060 – Infravec2). Infravec2 sponsored the RNA sequencing at the Polo GGB facility in Siena, Italy and bioinformatics support at FORTH IMBB, Greece. The Inferavec2 grant was awarded to NW (Grant No. 5505). The funders had no role in study design, data collection and analysis, decision to publish, or preparation of the manuscript.

**Competing interests:** The authors have declared that no competing interests exist.

## Author summary

Insecticide-based mosquito control has saved millions of lives from malaria and other vector-borne diseases. However, the emergence and increase of insecticide resistant *Anopheles* populations seriously threaten to derail malaria control programmes. Surveillance of insecticide resistance and understanding the underlying molecular mechanisms are key for choosing effective vector control strategies. Here, we characterised the degree and mechanisms of resistance in three malaria vector populations from Côte d'Ivoire. Our key finding was that these multi-insecticide resistant malaria vectors largely remained susceptible to malathion, despite the presence of a mutation in the target enzyme of this organophosphate insecticide that would typically confer resistance. Intriguingly, we found overexpression of metabolic P450 enzymes that are known to detoxify insecticides and activate pro-insecticides such as malathion. It is highly probable that, here, we observed P450-mediated negative cross-resistance for the first time in *Anopheles* field populations. Negative cross-resistance merits further investigation as advantage could be taken of this phenomenon in the fight against multi-resistant malaria vectors.

## Introduction

Insecticide-based vector control interventions – primarily insecticide-treated bed nets (ITNs) and indoor residual spraying (IRS) – have saved millions of lives from malaria and other vector-borne diseases across Africa [1,2]. However, the extensive use of a limited arsenal of insecticides, applied not only in vector control but also in agriculture, has imposed an immense selection pressure on malaria vectors. The resulting emergence and rapid spread of insecticide resistant *Anopheles* mosquitoes are now seriously threatening the success of malaria control efforts [3,4]. Indeed, insecticide resistance is regarded an important contributor to the stalling progress of malaria prevention and control [2].

A crucial challenge is to maintain the efficacy of current interventions by delaying and managing insecticide resistance as well as bringing new control tools to the market. Surveillance of insecticide susceptibility in mosquito populations is key to guide vector control strategies in order to make them as effective and sustainable as possible [5]. Phenotypic insecticide resistance is detected by exposing live mosquitoes to insecticides in susceptibility bioassays and subsequently observing mortality rates. In addition, understanding the underlying genomic alterations helps identifying the risk for cross-resistance and is thus fundamental to define the most potent insecticide resistance management (IRM) strategy in a given setting [5,6].

The main physiological resistance mechanisms described in the major African malaria vectors are i) modification of the insecticide target site; ii) enhanced sequestration, detoxification or excretion of insecticides; and iii) decreased insecticide penetration through cuticular alterations [7,8]. Once genomic markers for resistance mechanisms are identified, they can be implemented in vector diagnostics in the frame of entomological surveillance and will support IRM decision-making.

Reduced target-site sensitivity is caused by mutations leading to amino acid substitutions that hamper insecticide binding. Two point mutations at the same codon position of the voltage-gated sodium channel (Vgsc) in *Anopheles gambiae* s.l., *Vgsc*-L995F/S (previously referred to as *Vgsc*-L1014F/S), induce *knockdown resistance* (*kdr*) to pyrethroids (PYs) and organochlorines (OCs) [9,10]. Both insecticide classes target the Vgsc in the insect's neuronal cell membrane. A third mutation, *Vgsc*-N1570Y (previously referred to as *Vgsc*-N1575Y), enhances the *kdr* phenotype of mosquitoes that already carry the mutant 995F allele and thus is called

super-*kdr* mutation [11]. Advances in whole mosquito genome sequencing reveal more and more non-synonymous mutations associated with insecticide resistance, for example, *Vgsc*-I1527T [12–14]. The acetylcholinesterase mutation *Ace1*-G280S (previously referred to as *Ace1*-G119S) confers cross-resistance to carbamates (CBs) and organophosphates (OPs) that target the synaptic enzyme [15]. Recent studies have discovered how the complex architecture of the *Ace1* locus with its heterogeneous and homogeneous duplications elevates resistance levels while compensating for the disadvantageous fitness consequences [16,17].

Metabolic resistance is caused by enhanced detoxification of insecticides through increased biodegradation, sequestration or excretion. Three major gene families encode for the key enzymes implicated in metabolic resistance: carboxylesterases (COEs), glutathione *S*-transferases (GSTs) and cytochrome P450-dependent monooxygenases (P450s) [8]. Additional gene families may be involved in insecticide detoxification, such as ATP-binding cassette (ABC) transporters [18], chemosensory proteins (CSP) [19] and uridine diphosphate (UDP)-glycosyltransferases (UGTs) [20]. Many members of these six enzyme families were found to be upregulated in a variety of microarray experiments comparing insecticide-resistant with -susceptible mosquitoes [21,22], but only a handful have been functionally validated in follow-up experiments. Examples are *CYP6P3* [23] and *CYP6M2* [24], which were both validated by heterologous expression in *Escherichia coli*, confirming that these enzymes can metabolise a range of insecticides including several PYs and OPs *in vitro* [25].

The prevalence of insecticide resistance in malaria vector populations is increasing across Africa with extreme resistance phenotypes being reported from West Africa [26,27], including Côte d'Ivoire [28]. Côte d'Ivoire has a long history of research on insecticide resistance with the first reports of PY resistance dating back to 1993 [29]. Since then, the prevalence of resistance has been steadily increasing for different insecticides and across regions [30,31]. Previous studies by Edi and colleagues on *An. coluzzii* mosquitoes from Tiassalé, a town in southern Côte d'Ivoire, have shown that increased insecticide metabolism by overexpressed CYP6 P450s alongside *kdr* target-site mutations, and duplications of the mutant *Ace1* locus, contribute to multiple insecticide resistance phenotypes across four insecticide classes (i.e. PY, OC, CB and OP) [32,33]. A study, assessing insecticide susceptibility in *An. gambiae* s.l. mosquitoes from ten Ivorian sites using the World Health Organization (WHO) insecticide susceptibility test, found all populations to be resistant to the PY deltamethrin, the OC dichlorodiphenyltrichloroethane (DDT) and the CB bendiocarb, while malathion (OP) susceptibility varied between sites [34].

A possible explanation for the observed resistance pattern in Côte d'Ivoire is negative cross-resistance, whereby the same molecular mechanism conferring resistance to one insecticide simultaneously induces increased susceptibility to another insecticide. Perhaps this could be the case for malathion, in that overexpression of P450s conferring resistance to PYs increases the susceptibility to OPs by producing metabolites that are even more toxic [35]. Interestingly, malathion is a pro-insecticide that is activated to the more active malaoxon through P450-mediated oxidative reactions. While some key P450s belonging to the CYP6 and CYP9 family have been shown *in vitro* to metabolise insecticides from multiple classes, including OPs [25], CYP6M2 overexpression has recently been postulated to induce negative cross-resistance *in vivo* [36]. However, specific loci associated with negative cross-resistance in field populations are yet to be elucidated. Such markers could help identifying which insecticides are more toxic to multi-resistant mosquitoes compared to their more susceptible counterparts, preserving the massive achievements gained in the fight against malaria.

In this study, we measured the degree of phenotypic insecticide resistance against deltamethrin and malathion using dose-response bioassays of three *An. coluzzii* field populations from southern Côte d'Ivoire. Additionally, we investigated the underlying molecular mechanisms

for the observed phenotypes by RNA sequencing (RNA-seq) and quantitative PCR (qPCR) in individual mosquito specimens.

## Materials and methods

### Field sites and mosquito populations

*Anopheles gambiae* s.l. field populations were collected at the larval stage in three intensively cultivated agricultural sites in southern Côte d'Ivoire, including Agboville (Agb, Latitude: 5.945129˚, Longitude: -4.235667˚), Dabou (Dab, 5.318682˚, -4.264588˚) and Tiassalé (Tia, 5.890383˚, -4.831223˚) (Fig 1). In Tiassalé and Agboville, we collected *Anopheles* larvae from irrigated rice fields, whereas the collected larvae in Dabou were from standing water in vegetable fields. We carried out larval collections in June and July 2018, when the climate is hot and rainy in this area of West Africa. We reared the larvae to the adult stage by feeding them ground Tetramin fish food (Tetra, Melle, Germany) in the insectary of the Centre Suisse de Recherches Scientifiques en Côte d'Ivoire (CSRS) in Abidjan. We fed the adult mosquitoes with 10% sucrose solution and kept females and males in the same cages, allowing them to mate. The conditions for rearing and testing were standardised at 26˚C ± 2˚C, with 75% ± 10% relative humidity and a 12:12 h photoperiod.

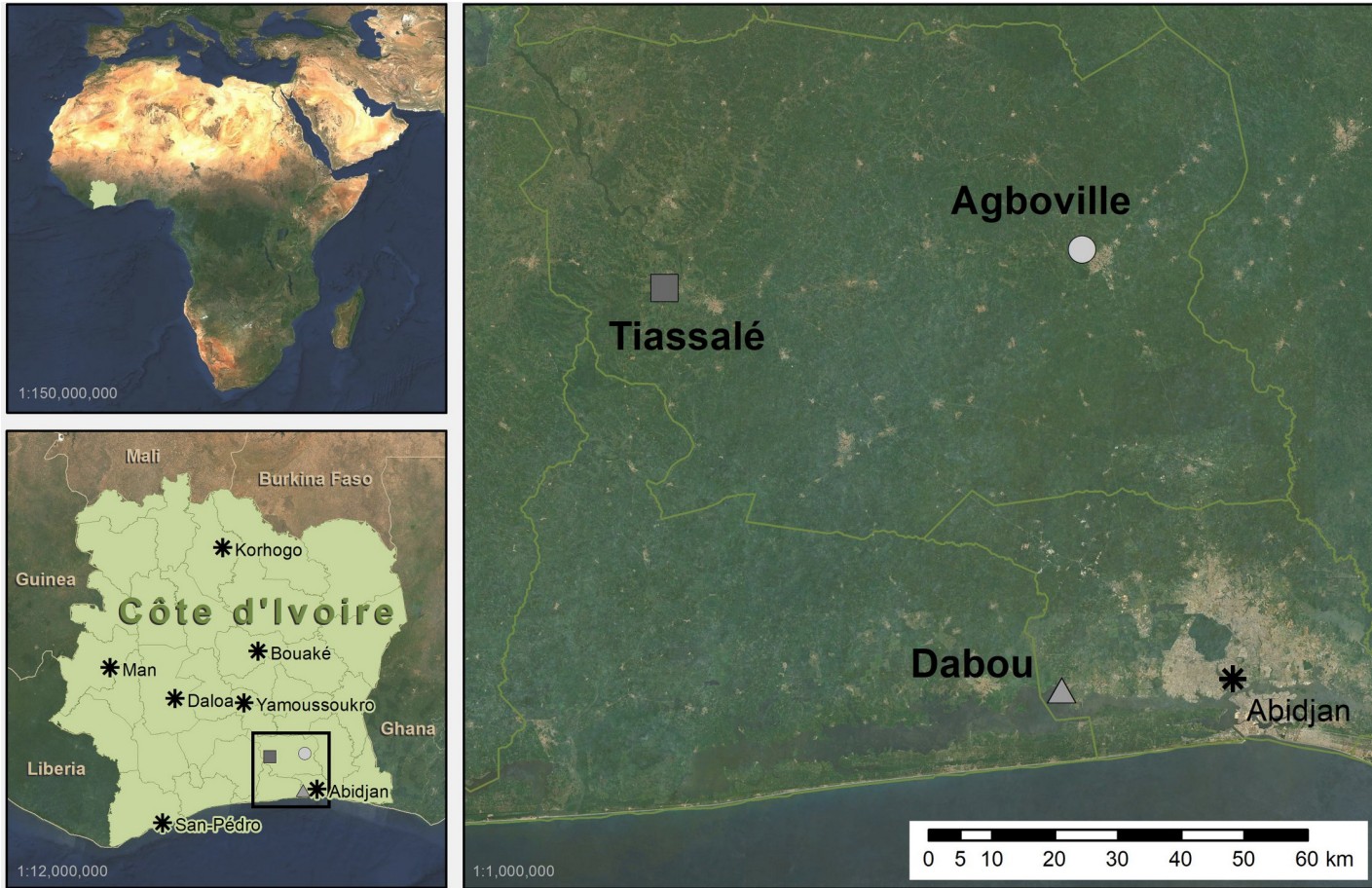

**Fig 1. Map of the three larval sampling sites in southern Côte d'Ivoire.** In Tiassalé and Agboville we collected *Anopheles* larvae from irrigated rice fields, whereas in Dabou they were collected from standing water in vegetable fields. The map was created using ArcGIS v10.6.1 (ESRI Inc., Redlands, CA, USA) with the base map Sentinel-2 cloudless - https://s2maps.eu by EOX IT Services GmbH (contains modified Copernicus Sentinel data 2020).

We used the laboratory strain Ngousso, colonised in the Molecular Entomology laboratory at FORTH-IMBB, Greece, as an insecticide susceptible comparator population for both the phenotypic bioassays and molecular analyses. The *An. coluzzii* Ngousso colony originates from a neighbourhood called "Ngousso" in Yaoundé, Cameroon, and was established in 2006 [37]. As an additional susceptible comparator population for the molecular analyses, we ordered eggs of the Mali-NIH (MRA-860) strain from BEI resources and raised them to the adult stage at the Swiss Tropical and Public Health Institute (Swiss TPH). The *An. coluzzii* Mali-NIH strain originates from Niono, a town in central Mali, and was established in 2005. We reared and tested the susceptible colonies under the same conditions as described above for the field populations.

## Bioassays

We evaluated the field populations' phenotypic resistance to four public health insecticides recommended by WHO. We tested the susceptibility to discriminating concentrations of bendiocarb (0.1%), deltamethrin (0.05%), DDT (4%) and malathion (5%) [6]. Additionally, we established dose-response curves with serial dilutions of deltamethrin and malathion to assess the resistance intensity of the three field populations compared to the susceptible Ngousso colony.

We prepared insecticide treated filter papers according to the WHO guidelines for testing mosquito adulticides [38]. We weighed the calculated amount of analytical grade insecticide (all from PESTANAL, Sigma-Aldrich, Saint Louis, MO, USA) on a precision balance and dissolved it in the appropriate volume of a freshly prepared mixture of solvent (acetone) and carrier oil. Silicon oil (DOWSIL 556 cosmetic grade fluid) served as carrier for deltamethrin and DDT, while olive oil (Sigma-Aldrich, Cat. No. 75343) was used for malathion and bendiocarb. According to the WHO guidelines, one filter paper has to be impregnated with 2 ml of the mixture of volatile solvent and 648 mg carrier oil. As the needed amount of carrier depends on the oil's density (in $g/cm^3$), the mixture ratios per paper were 0.66 ml silicon oil (0.98 $g/cm^3$) with 1.34 ml acetone and 0.71 ml olive oil (0.91 $g/cm^3$) with 1.29 ml acetone. Insecticide concentrations are expressed as the percentage of active ingredient per unit volume of non-volatile carrier oil on the filter paper. Using a 10 ml glass pipette, we evenly pipetted 2 ml of insecticide-oil-acetone mixture onto a 12 cm x 15 cm Whatman qualitative filter paper, Grade 1 (Sigma-Aldrich, Cat. No. 1001917) placed on a support rack made of a cardboard pierced with 30–40 nails. For each prepared filter paper batch, we treated a set of control papers with 2 ml of the corresponding carrier oil-acetone mixture without insecticide. We let the impregnated papers air-dry for 24 h, protected from light in a chemical hood. We wrapped the papers in aluminium foil, placed them in zip-lock plastic bags and stored them at 4˚C. We used each paper for a maximum of six times within 10 weeks after impregnation.

For the bioassays, we exposed batches of 20–25 mosquitoes for 1 h to the insecticide treated filter papers inserted into WHO plastic tubes following the WHO test procedures [6]. We used non-blood-fed, 2- to 5-day-old adult females for all bioassays and subsequent molecular analyses. The adults were either individuals hatching from the field-collected larvae or from the susceptible laboratory colonies. After 1 h insecticide exposure, the mosquitoes were gently blown back to the holding tube lined with a clean paper and provided with 10% sucrose solution. After a 24 h recovery period, we counted the numbers of dead and alive mosquitoes in each tube. To establish the dose-mortality curves, we tested 5–8 different malathion and deltamethrin concentrations that resulted between 0% and 100% mortality for a given population. We repeated the test until we exposed at least 100 mosquitoes per concentration and population. We did not have sufficient numbers of the susceptible Mali-NIH colony to perform conclusive

dose-response assays. In each bioassay round, we included 25–50 unexposed control mosquitoes subjected to the same experimental procedures except that they were exposed to control papers treated with solvent and oil only.

## Experimental design

The molecular analyses included five laboratory-reared *An. coluzzii* populations: three multi-resistant populations collected at the larval stage in Côte d'Ivoire (Agb, Dab and Tia) (Fig 1) and two insecticide susceptible laboratory colonies (Lab2: Mali-NIH and Ngousso). We analysed mosquitoes that were alive 24 h post-exposure to three experimental conditions: unexposed to insecticide, meaning exposed to a control paper (C), exposed to 6.4% deltamethrin (D) and exposed to 2.5% malathion (M). We included unexposed control specimens from all five populations, while we included surviving insecticide-exposed specimens only from the three Ivorian populations, thereby selecting the most resistant females of each field population per insecticide. While we aimed at selecting the 25% most resistant mosquitoes, the actual percentage varied slightly between the three field populations due to differences in the resistance phenotype. For protocol consistency, we decided to work with the same insecticide concentrations for all of them. We randomly chose individuals per experimental condition and population to be molecularly tested using the base R function *sample()* [39]. We subjected DNA of 40 unexposed and five selected individual females for each insecticide to qPCRs for species identification and detection of the four target-site mutations *Vgsc*-L995F/S, *Vgsc*-N1570Y and *Ace1*-G280S. In addition, we tested five individuals from each susceptible population with the same species identification and target-site mutation assays. We subjected RNA of the same five individuals, meaning five biological replicates (not pooled mosquitoes), per population and experimental condition to RNA-seq. See S1 Fig depicting the RNA-seq experimental design. Additionally, we subjected RNA of the same five individuals per experimental condition and population to reverse transcription (RT-)qPCRs for gene expression analysis of eight genes implicated in metabolic insecticide resistance; GSTE2, CYP6P3, CYP6M2, CYP9K1, CYP6P4, CYP6Z1, CYP6P1 and CYP4G16.

## Sample preservation and nucleic acid extractions

After completing the bioassays with recording mortality 24 h post-exposure, we preserved all mosquitoes in RNA*later* (Ambion, Inc., Austin, Texas, US, purchased from Sigma-Aldrich, Cat. No. R0901), separated by experimental condition (concentration of insecticide or control) and survival status (dead or alive). We mouth-aspirated alive mosquitoes and blew them into absolute ethanol to kill them. We also moistened the dead mosquitoes with absolute ethanol and blotted away the excess with a paper towel to make the tissue more permeable to RNA*later*.

We removed 1–3 legs from each individual while briefly placing the RNA*later* preserved mosquito on a chill table (BioQuip Products, Inc., Rancho Dominguez, CA, US, Cat. No. 1429). Genomic DNA was extracted from mosquito legs to identify the species and four insecticide target-site mutations with qPCR of both the field-collected and the susceptible laboratory populations. Total RNA was extracted from the remaining mosquito parts and used for RNA-seq and for RT-qPCRs to determine differential expression of eight metabolic resistance loci specified in the quantitative PCR section below.

For species identification and target-site mutation genotyping, we ground the mosquito legs in 100 μl DNAzol Reagent (Molecular Research Center, Inc., purchased from Thermo Fisher Scientific, Waltham, MA, USA, Cat. No. 10503027) using a battery-powered tissue grinder (Sigma-Aldrich, Cat. No. Z359971) and a plastic pestle in a 1.5 ml micro centrifuge

tube. We followed the manufacturer's protocol with the following modifications. Along with 50 μl of absolute ethanol for DNA precipitation, we added 10 μg of glycogen (Roche Diagnostics, Rotkreuz, Switzerland, Cat. No. 10901393001) to increase DNA yield. To pellet the precipitated DNA, we centrifuged the sample at 10,000 ×*g* for 12 min at 4˚C. After washing the pellet twice with 500 μl of 75% ethanol and centrifuging for 5 min at 10,000 ×*g*, we let the samples air-dry for 15 min on a heat-block set at 50˚C. Finally, we solubilised the DNA in 15 μl DNA/DNase/RNase-free water and stored the samples at -20˚C until qPCR analyses.

After amputating 1–3 legs, we isolated the total RNA from RNA*later*-preserved, individual mosquitoes using TRI Reagent (Molecular Research Center, Inc., purchased from Sigma-Aldrich, Cat. No. T9424). After mechanical grinding in 70 μl TRI Reagent as described above for DNA extraction, we added an additional 430 μl TRI Reagent to the sample and used it to rinse the pestle at the same time. We followed the manufacturer's protocol and opted for 50 μl of 1-bromo-3-chloropropane for the phase separation. After two washes with 75% ethanol, we solubilised the air-dried RNA pellets in 30 μl DNA/DNase/RNase-free water. To remove genomic DNA contamination, we treated each sample with the TURBO DNA-*free* Kit (Invitrogen, Cat. No. AM1907) according to the manufacturer's protocol. We shock-froze the RNA samples in liquid nitrogen and stored them at -80˚C. For RNA quantification, we measured RNA concentrations using the Qubit RNA High Sensitivity assay kit (Invitrogen, Cat. No. Q32852) on a Qubit 4 Fluorometer with 1 μl sample input. For RNA quality control, we used the Fragment Analyzer High Sensitivity RNA Analysis Kit (Agilent Technologies, Santa Clara, CA, US, Cat. No. DNF-472) with 2 μl sample input. For additional quality control, we visualised a random subset of RNA samples (200 ng/sample) on a 1.2% non-denaturing agarose gel stained with GelRed (Biotium, Fermont, CA, USA).

### RNA sequencing

The sequencing facility Polo-GGB in Siena, Italy, performed the preparation of RNA-seq libraries and sequencing. For each of the 55 individual laboratory-reared, 3- to 6-day-old *An. coluzzii* females, a library was prepared using the Illumina TruSeq Stranded mRNA Kit according to the reference guide (1000000040498 v00). In brief, after purifying and fragmenting poly (A)+ messenger RNA (mRNA), double-stranded complementary DNA (ds cDNA) was generated. Index adapters were ligated to the ds cDNA and those fragments with adapters on both ends were amplified. Following library validation on the Fragment Analyzer and quantification using Qubit, indexed libraries were normalised to 4 nM and pooled for sequencing. Paired-end sequencing of 75 nucleotides (PE 2×75) was carried out on an Illumina NextSeq 550 Instrument using the Illumina NextSeq 500/550 v2.5 High-Output chemistry and 1% of Phix control. The same pool of 55 indexed libraries was sequenced twice on different flow cells, thus producing two technical replicates per biological sample in two sequencing runs.

### Quantitative PCR assays for species identification, metabolic resistance loci and target-site mutations

For all qPCR assays, we used TaqMan probes labelled at the 5' end with the fluorophores FAM or HEX in the duplex or FAM, HEX or Atto647N in the triplex assays. All primer and probe sequences with their 5' and 3' modifications, concentrations and the suppliers are listed in S1A and S1B Table.

### Species identification qPCR assays

To make sure we only proceeded with *An. coluzzii* individuals, we analysed DNA from mosquito legs with two complementary qPCRs for species identification. To distinguish *An.*

*coluzzii/gambiae* s.s. (Ag+) as a group from *An. bwambae/melas/merus/quadirannulatus* (Aq+) as a group and from *An. arabiensis* (Aa+) we used common primers designed by Walker et al. [40] with species- or group-specific probes designed by Bass et al. [41]. We shortened the Ag+ and Aq+ probes by three nucleotides each at the 5' end to decrease unspecific binding. To additionally increase mismatch sensitivity, we added a minor groove binder (MGB) at the 3' end of all three probes.

To then distinguish *An. coluzzii* (formerly called molecular M-form) from *An. gambiae* s.s. (formerly called molecular S-form) in Ag+ samples, we used a newly developed TaqMan assay. We used the common primers published by Santolamazza et al. [42] that flank a 200 bp short interspersed element (SINE200) on the X chromosome. This SINE200 locus, called S200 X6.1, is fixed in *An. coluzzii* but absent in *An. gambiae* s.s. specimens. We designed two probes within a genomic region flanked by the S200 X6.1 primers: the *An. coluzzii*-specific HEX-probe targeted the S200 X6.1 insertion, while the *An. gambiae* s.s./*arabiensis*-specific FAM-probe was designed over the junction of this insertion and also includes a T/C nucleotide substitution specific for these two species.

## Metabolic resistance gene expression RT-qPCR assays

We used the RT-qPCR assays developed by Mavridis et al. [43] on RNA extracted from individuals to measure metabolic resistance potentially conferred by the glutathione *S*-transferase GSTE2 and the cytochrome P450-dependent monooxygenases CYP6P3, CYP6M2, CYP9K1, CYP6P4, CYP6Z1, CYP6P1 and CYP4G16. We measured relative gene expression with an internal control locus–the housekeeping gene encoding the ribosomal protein S7 (RPS7)–as calibrator in each triplex reaction. We calculated expression levels of the field populations relative to the reference laboratory colonies according to Pfaffl et al. [44], implemented in the REST 2009 software v2.0.13 [45].

## Target-site mutation qPCR assays

We further tested mosquito leg DNA for four point mutations at insecticide target sites that have been previously associated with insecticide resistance. To detect two *kdr* mutations that lead to *Vgsc*-L995F/S substitutions in the PY and DDT target site, the voltage-gated sodium channel, we used the TaqMan assays designed by Bass et al. [46] in the triplex format optimised by Mavridis et al. [47]. To detect the super-*kdr* mutation *Vgsc*-N1570Y that enhances the 995F phenotype, we used the duplex TaqMan assay designed by Jones et al. [11]. We further tested for the mutation that leads to the *Ace1*-G280S substitution making the acetylcholinesterase enzyme less sensitive to CBs and OPs [48].

## General qPCR assay conditions

For all one-step (RT-)qPCR assays, we used 1 μl DNA or RNA extracted from individual mosquitoes in a total reaction volume of 10 μl with a mastermix supplied by Fast-Track Diagnostics (FTD, Esch-sur-Alzette, Luxembourg) and the primer and probe concentrations given in S1A Table. Reactions were performed in 96-well plates (Sarstedt, Cat. No. 72.1980.202) on a C1000 Touch CFX96 Real-Time PCR System (Bio-Rad Laboratories, Hercules, CA, US). We used the same thermal cycle parameters for all assays: reverse transcription at 50˚C for 15 min, RTase inactivation and initial denaturation at 95˚C for 3 min, followed by 40 cycles of denaturation at 95˚C for 3 s and annealing and extension at 60˚C for 30 s. For each biological sample, we ran technical duplicates of each qPCR reaction and included positive and non-template controls for each assay and on every 96-well plate. For qPCR data management and quality control we used the web-based, open-source platform ELIMU-MDx [49].

## Data analysis

We statistically analysed and visualised the data using the freely available software R, version 3.6.3 [39] in the integrated development environment RStudio [50]. For data tidying and visualisation, we used the *tidyverse* packages [51] and for fitting the generalised linear models (GLMs), we used the *lme4* package [52].

## Bioassay data analysis

To assess the phenotypic insecticide resistance status we followed the WHO protocol for insecticide susceptibility tests with discriminating concentrations [6]. We estimated the confidence intervals for each insecticide based on GLMs. We fitted the binary bioassay outcome *dead* or *alive* using a GLM with a binomial distribution and a logit link function and included a term for *field site* into the model.

To estimate the dose-response relationship between 24 h mortality and insecticide concentration we exposed the mosquitoes to a range of different concentrations of deltamethrin and malathion. With the aim to predict mortality as a function of the log-transformed insecticide concentration, we fitted the binary *dead* or *alive* outcome with a GLM with a binomial distribution and a logit link function as previously described by Suter et al. [53]. The estimated dose-response curves allowed us to determine the lethal concentration (LC) that would kill 50% ($LC_{50}$) of the different mosquito field populations and the comparator laboratory colony Ngousso. Resistance ratios ($RR_{50}$) were calculated by dividing the respective $LC_{50}$ of each field populations by the Ngousso $LC_{50}$.

## RNA sequencing data analysis

To demultiplex the RNA-seq data we assigned the sequenced reads to each barcoded sample and trimmed off the indices using the Illumina bcl2fastq v2.20.0.422 tool with the *–no-lane-splitting* option. We checked the read quality of each sample applying the programs FastQC [54] and MultiQC [55]. Using STAR [56] we aligned the reads to the *An. coluzzii* Ngousso reference genome (Acol N1.0) that includes 13,299 genes and is available on VectorBase [57,58]. Subsequently we quantified gene-level expression using featureCount with the strand-specific read counting option [59]. We used the freely available Bioconductor package *edgeR* for the differential gene expression analysis [60]. We filtered out genes with very low read counts following the strategy from Chen et al. [61], as implemented in *edgeR*, with default parameters. Then we normalised library sizes using the trimmed mean of M-values (TMM) method [62] that is part of edgeR. We used the GLM approach to build several contrasts for differential gene expression analysis. To test for differences in gene expression between the three Ivorian *An. coluzzii* populations, we compared the insecticide-unexposed controls of the field populations between each other (Agb_C vs. Dab_C, Agb_C vs. Tia_C, Dab_C vs. Tia_C). To further investigate the effect of insecticide selection within field populations, we compared the malathion or deltamethrin survivors vs. the corresponding control of the same field population and also malathion vs. deltamethrin survivors (Agb_D vs. Agb_C, Agb_M vs. Agb_C, Agb_D vs. Agb_M, etc.). Finally, with the aim to find genes that are possibly involved in insecticide resistance and constitutively differentially expressed between multi-resistant field populations and susceptible laboratory colonies, we compared unexposed control specimens of each field site against the two susceptible colonies as a group (Agb_C vs. Lab2_C, Dab_C vs. Lab2_C, Tia_C vs. Lab2_C). The p-values were adjusted applying the Benjamini and Hochberg procedure to control for the False Discovery Rate (FDR) [63]. We defined the cut-off for overexpressed genes at $\log_2$ fold change ($\log_2$FC) $> 0$ and FDR $\leq 0.01$ and for underexpressed genes at $\log_2$FC $< 0$ and FDR $\leq 0.01$.

With the aim to complement the currently sparsely annotated *An. coluzzii* genome (ACON), we downloaded corresponding orthologues from six other members of the *An. gambiae* complex (AARA, ACOM, AGAP, AMEC, AMEM and AQUA) available on BioMart [64]. Moreover, we verified our merged annotation list by comparing it to the data provided in recent publications on insecticide resistance in malaria vectors and adopted additional annotations including three members of the *Maf-S-cnc-Keap1* pathway and 17 UGTs [65,66]. In addition, we performed a Blastp [67] search of all *An. coluzzii* predicted proteins (n = 14,694) against the SwissProt section of UniProt [68] (n = 563,972), and added the annotation of the highest-scoring hit from SwissProt to an *An. coluzzii* protein when the expectancy E-value was below $10^{-10}$. The complete annotation list is reported in S2A Table.

We tested the enrichment of specific gene sets including protein families whose functions could be related to insecticide resistance (sets in S2A Table). For this purpose, we used the gene set enrichment analysis (GSEA) method implemented in the R package *fgsea* [69] with $\log_2$FC as a metric and Fisher's exact test on the numbers of significantly differentially expressed genes (DEGs). After using the Benjamini-Hochberg procedure to correct for multiple testing [63], we regarded a gene set as significantly enriched only when the FDR of both the GSEA and Fisher's exact test was $\leq 0.01$.

We performed joint variant calling on the RNA-seq data using the Genome Analysis Toolkit (GATK) HaplotypeCaller [70]. Genotypes were inferred on a position when the PHRED probability of having at least one variant allele was 20 (*-stand-call-conf*). Furthermore, individual inferred genotypes were kept when their PHRED quality was at least 10 and the coverage at least 3 at that position for that sample.

## Results

### Phenotypic insecticide resistance

We collected *An. gambiae* s.l. larvae in Agboville, Dabou and Tiassalé in southern Côte d'Ivoire (Fig 1) and reared them to adults (n = 6,536 females phenotyped in bioassays) under standardised laboratory conditions. With the exception of one individual from Tiassalé that turned out to be an *An. coluzzii/gambiae* s.s. hybrid, all molecularly tested specimens (n = 222) were identified as *An. coluzzii*. Therefore, we considered the populations to be *An. coluzzii*.

To assess the insecticide resistance phenotypes, we tested 2- to 5-day-old adult females in WHO insecticide susceptibility tests using WHO discriminating concentrations [6]. All three *An. gambiae* s.l. field populations were resistant to bendiocarb, DDT and deltamethrin (Fig 2A). In contrast, the Tiassalé population was fully susceptible to malathion and, while possible resistance may be suspected for the Agboville population and resistance was shown for the Dabou population, the $LC_{50}$ resistance ratios between the three field populations and the laboratory susceptible Ngousso reference colony were very low (see below) (Fig 2A and 2C and Table 1).

To quantify the resistance levels we adapted the WHO insecticide susceptibility bioassay by recording mortality across a series of deltamethrin and malathion concentrations to determine the $LC_{50}$s of the mosquito populations based on the estimated dose-response curves shown in Fig 2B and 2C. Using the $LC_{50}$ of the susceptible Ngousso colony as the denominator, the resistance ratios ($RR_{50}$) for deltamethrin were 388.5-fold for Agboville, 1,134.4-fold for Dabou and 1,802.9-fold for Tiassalé (Table 1). At the same time, we observed much lower $RR_{50}$s for malathion with 5.9-fold for Agboville, 8.6-fold for Dabou and 2.6-fold for Tiassalé. In summary, dose-response bioassays revealed extremely high deltamethrin resistance in all sites, with the highest $RR_{50}$ in Tiassalé. In contrast, the $RR_{50}$s for malathion seem negligible, with the lowest $RR_{50}$ in Tiassalé, affirming this population was still the most susceptible to malathion.

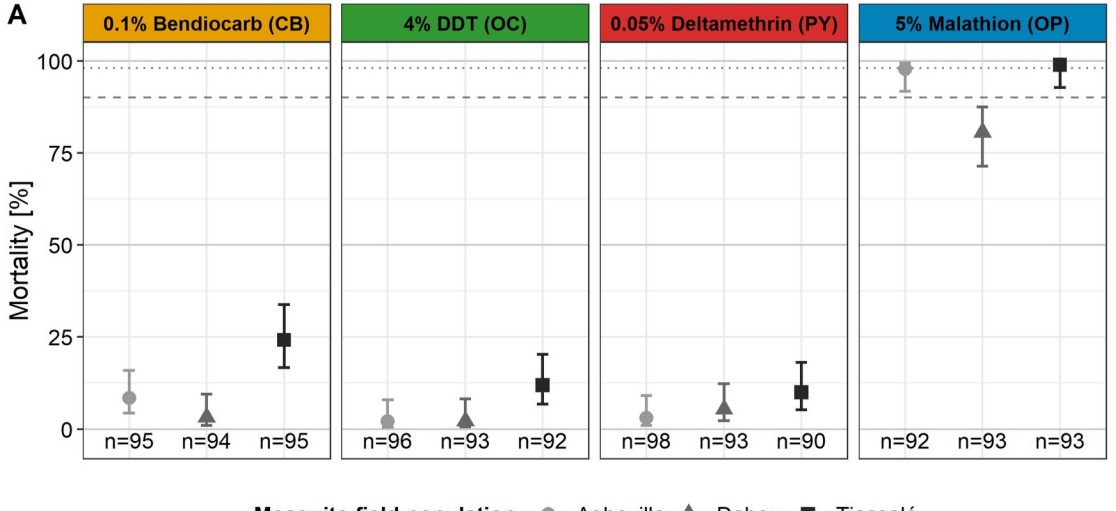

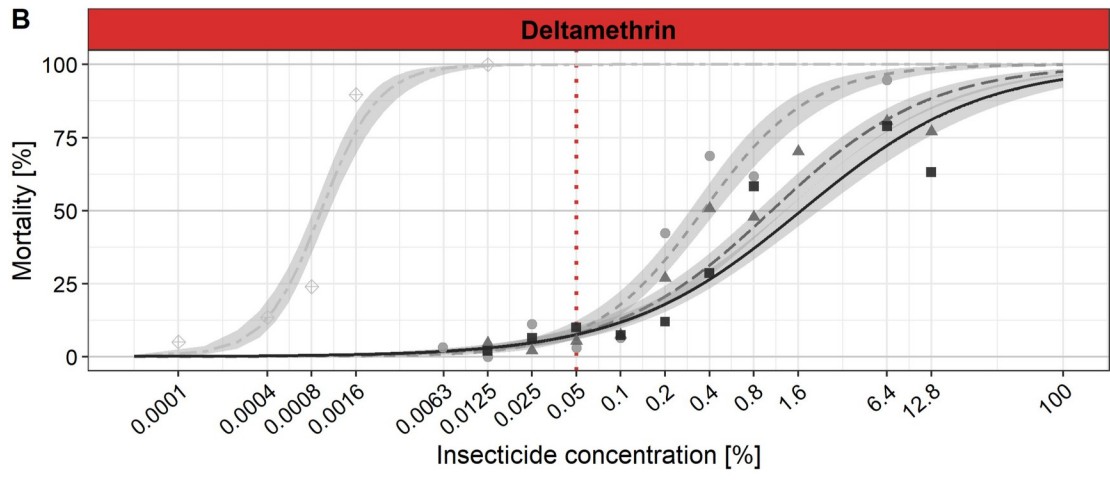

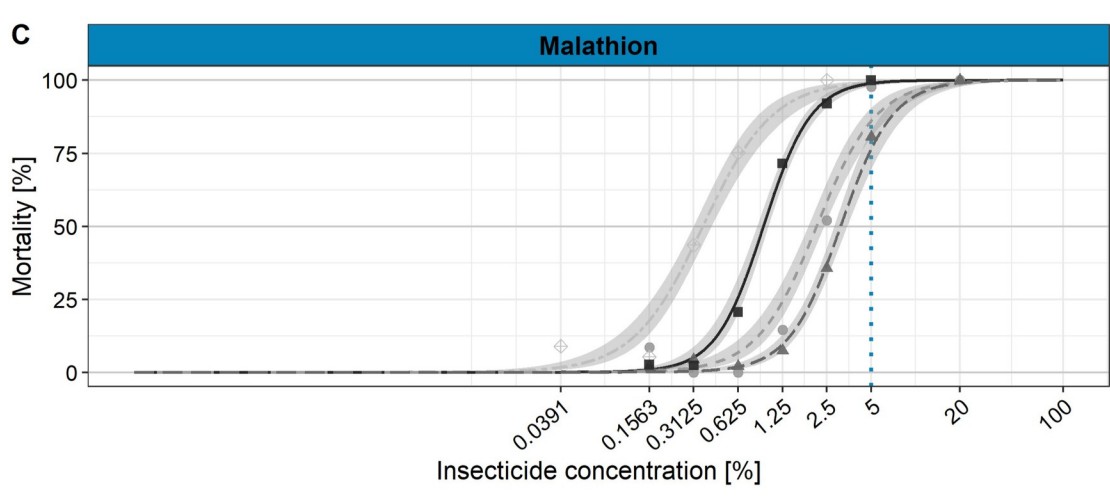

**Fig 2. Phenotypic insecticide resistance in *Anopheles coluzzii* from southern Côte d'Ivoire assessed with WHO discriminating concentrations and dose-response bioassays. (A)** Outcome of the standard WHO insecticide susceptibility tests using discriminating concentrations of four insecticides. The solid symbols represent the average mortalities and the vertical solid lines with whiskers represent their 95% confidence intervals (CIs) as estimated with GLMs. Note that for the Tiassalé population we observed 100% mortality 24 h post malathion exposure, but in order to estimate a 95% CI, we artificially added one survivor into the model. The total number (n) of tested mosquitoes per field site and insecticide is indicated below the corresponding average mortality. Horizontal dashed lines represent thresholds to interpret the WHO susceptibility tests [6]: i) A mosquito population is susceptible to the tested insecticide if the observed mortality is between 98% and 100% (above upper line); ii) if mortality is between 90% and 97% resistance is possible (between lines); and iii) if mortality is below 90%, resistance is confirmed (below lower line). Dose-response curves for **(B)** deltamethrin and **(C)** malathion for the three *An. coluzzii* field populations from southern Côte d'Ivoire and the insecticide susceptible Ngousso laboratory colony. The curves show the dose-response relationship between the mortality rate on the *y*-axis and the percentage of insecticide on the filter paper on the log$_2$-transformed *x*-axis. Symbols represent the actually observed mortality rates per dose, while the curves are the estimates of the average as estimated with GLMs. The shaded areas indicate the 95% CIs. Vertical lines indicate the discriminating concentration for insecticide susceptibility as recommended by the WHO.

## RNA sequencing results

We individually sequenced the transcriptomes of 55 laboratory-reared, 3- to 6-day-old *An. coluzzii* females using RNA-seq. Each mosquito was sequenced twice in different runs, producing two technical replicates. Five individuals (i.e. five biological replicates) per population and experimental condition were sequenced (S1 Fig). The five laboratory-reared *An. coluzzii* colonies were three multi-resistant populations collected at the larval stage in Côte d'Ivoire (Agb, Dab and Tia) and the two insecticide susceptible laboratory strains Mali-NIH and Ngousso (Lab2). The three experimental conditions were insecticide-unexposed control (C) for all populations as well as selected against 6.4% deltamethrin (D) and 2.5% malathion (M) for the Ivorian populations only.

We excluded one individual from the analysis, a deltamethrin survivor from Agboville (ID: 01_Agb_D02) because we obtained only ~2.5 million reads and low mapping rates (~30%) in both sequencing runs. The number of reads per run for the other 54 individuals ranged from ~6.6 to 11.6 million while 84% - 95% were uniquely mapped to the *An. coluzzii* reference genome.

Multi-dimensional scaling (MDS) plots indicated lower variation between runs (i.e. technical replicates) than between individuals (i.e. biological replicates) (S2A Fig). Thus, after controlling for potential batch effects, we summed the read counts from both runs to obtain the

**Table 1. Lethal concentrations and resistance ratios for deltamethrin and malathion in adult female *Anopheles coluzzii* from southern Côte d'Ivoire compared to the laboratory reference colony Ngousso.**

| Insecticide | Population | n [a] | LC$_{50}$ (95% CI) [b] | RR$_{50}$ [c] |
|---|---|---|---|---|
| Deltamethrin | Agboville (field) | 654 | 0.361 (0.293–0.445) | 388.5 |
| | Dabou (field) | 976 | 1.054 (0.848–1.310) | 1134.4 |
| | Tiassalé (field) | 1027 | 1.675 (1.283–2.188) | 1802.9 |
| | Ngousso (lab) | 486 | 0.001 (0.001–0.001) | 1 |
| Malathion | Agboville (field) | 561 | 2.128 (1.884–2.404) | 5.9 |
| | Dabou (field) | 818 | 3.098 (2.801–3.427) | 8.6 |
| | Tiassalé (field) | 762 | 0.928 (0.851–1.013) | 2.6 |
| | Ngousso (lab) | 482 | 0.360 (0.313–0.414) | 1 |

[a] Number (n) of mosquitoes tested per population and insecticide.

[b] Insecticide concentration (%) that is lethal for 50% (LC$_{50}$) of the mosquitoes. Concentrations are expressed as percentage of active ingredient per unit volume of non-volatile carrier oil on a filter paper to which mosquitoes were exposed to for 1h [6]. The 95% confidence intervals (CIs) were estimated with GLMs.

[c] Resistance ratio (RR$_{50}$) between the LC$_{50}$s of the Ivorian populations (field) and the insecticide susceptible laboratory reference colony Ngousso (LC$_{50(field)}$/ LC$_{50(lab)}$).

total number of reads per gene for each individual. Next, we filtered out genes with very low read counts and proceeded with 10,519 out of the 13,299 *An. coluzzii* genes for normalisation and subsequent differential gene expression analysis.

The gene expression patterns show clear separation between the two lab colonies and the field populations (S2A and S2B Fig) and also between the Mali-NIH and Ngousso lab colonies. In contrast, all field samples from Côte d'Ivoire group together without obvious patterns indicative of collection site or experimental condition (S2B and S2C Fig).

## Comparisons between and within multi-resistant populations from Côte d'Ivoire

Firstly, we compared the insecticide-unexposed field specimens of each site to each other to test for differences in gene expression between the three Ivorian *An. coluzzii* populations. Secondly, we compared specimens from the three different experimental conditions (unexposed control, malathion or deltamethrin survivors) within each field population to investigate if insecticide selection had an effect on gene expression. We found no or very few significantly differentially expressed genes (DEGs) – notably neither P450s nor any other genes typically associated with insecticide resistance – in these comparisons between or within the Ivorian *An. coluzzii* field populations (S3 Table).

## Comparison of multi-resistant Ivorian vs. susceptible laboratory populations

Secondly, we identified differentially expressed genes (DEGs) by comparing the number of reads between each multi-resistant field population (Agb, Dab or Tia) and the two insecticide susceptible laboratory colonies combined as a group (Lab2) (S4 Table). In these three comparisons, we only included control mosquitoes (C) that had been unexposed to insecticides with the aim to reveal genes that are constitutively over- or underexpressed between the field populations and the lab colonies. We found 395, 882 and 1,190 genes overexpressed ($\log_2$FC $> 0$, FDR $\leq 0.01$) while 397, 1,039 and 1,038 genes were underexpressed ($\log_2$FC $< 0$, FDR $\leq 0.01$) in Agboville, Dabou and Tiassalé, respectively, compared to the two laboratory colonies (Fig 3A–3D).

Among the 20 most up-regulated genes in Agboville were three carboxylesterases COEAE6G (FC = 40.7), COEAE8O (FC = 31.7) and COEAE6O (FC = 21.5) and two P450s CYP6P3 (FC = 23.2) and CYP6P5 (FC = 20.4). Further, the C-type lysozyme LYSC5 (FC = 25.3), two trypsins TRYP6 (FC = 36.4) and TRPY7 (FC = 31.9) and the cuticular protein CPLCP25 (FC = 68.5) (Fig 3A).

Among the 20 most up-regulated genes in Dabou were two carboxylesterases COEAE6G (FC = 88.2), COEAE3H (FC = 28.2) and the P450 CYP6P3 (FC = 31.7). We also found the anti-microbial peptide cecropin CEC4 (FC = 51.5) and three cuticular proteins CPLCP14 (FC = 55.9), CPR139 (FC = 38.6) and CPLCP25 (FC = 36.5) in the top 20 overexpressed genes in Dabou (Fig 3B).

Among the 20 most up-regulated genes in Tiassalé were the P450s CYP6P3 (FC = 28.1) and CYP6Z4 (FC = 21.4), and eight cuticular proteins with the highest fold change for CPLCP25 (FC = 101.7), followed by CPLCG5 (FC = 79.8), CPLCG4 (FC = 54.4), CPLCG3 (FC = 37.7), CPLCG2 (FC = 28.6), CPLCA3 (FC = 22.1), CPR125 (FC = 20.0) and CPR25 (19.0) (Fig 3C).

Far fewer genes were annotated through orthology among the top 20 down-regulated genes. In fact, for none of the top 20 down-regulated genes in the Dabou population we found a gene name based on orthology. In Agboville, the P450 CYP303A1 (FC = -200.3), the peroxidase HPX10 (FC = -21.8) and the CLIP-domain serine protease CLIPA5 (FC = -10.8) were

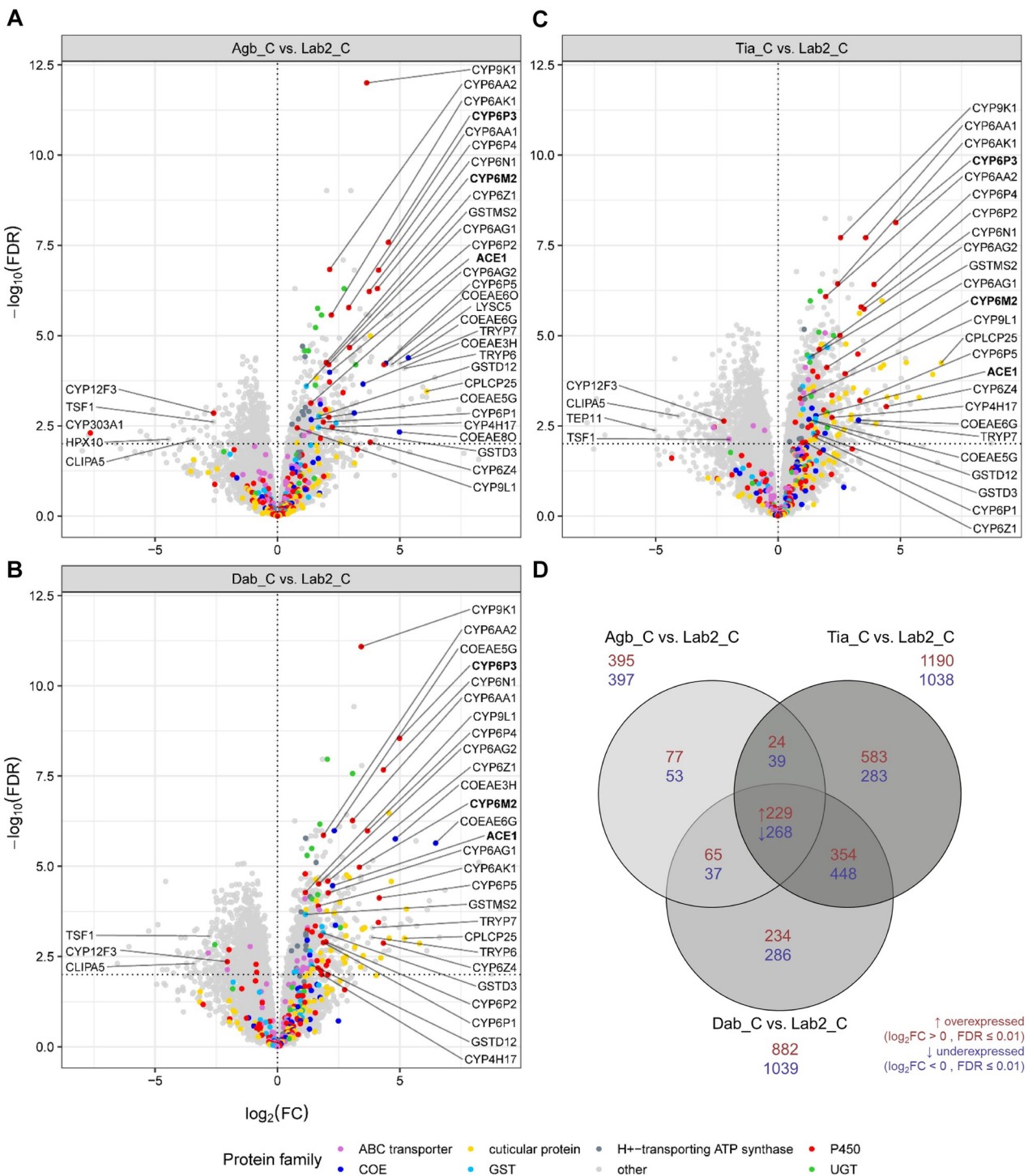

**Fig 3. Results of differential gene expression analysis comparing each multi-resistant field population (Agb_C, Dab_C and Tia_C) against two susceptible laboratory colonies combined as a group (Lab2_C: Mali-NIH and Ngousso).** Volcano plots showing the expression profile for comparisons **(A)** Agb_C vs. Lab2_C; **(B)** Dab_C vs. Lab2_C; and **(C)** Tia_C_ vs. Lab2_C. The genes highlighted in colour belong to protein families known to be involved in insecticide resistance and those that we detected significantly enriched with the GSEA for at least one site. Genes above the horizontal dotted line passed our threshold for significance (FDR $\leq$ 0.01). Genes on the right of the vertical line (log$_2$FC > 0) were higher expressed in the field than in the lab populations, whereas genes on the left of the vertical line (log$_2$FC < 0) were lower expressed in the field than in the lab populations. **(D)** Venn diagram showing the number of significantly (FDR $\leq$ 0.01) over- (log$_2$FC > 0, red) and under- (log$_2$FC < 0, blue) expressed genes.

down-regulated compared to the two lab colonies. The latter was also among the top 20 down-regulated genes in Tiassalé (CLIPA5, FC = -16.8) along with the thioester-containing protein TEP11 (FC = -32.8). Among the 268 genes that were commonly down-regulated in all three field sites, we had 11 annotated genes including the CYP12F3, a CLIP-domain serine protease (CLIPA5) and transferrin (TSF1) (Figs 3A–3C and S3A).

To detect the best candidates potentially involved in multiple insecticide resistance, we focussed on genes that were overexpressed three times independently in each field population compared to the susceptible colonies. Indeed, among these 229 commonly overexpressed genes, several belong to the three major enzyme families involved in metabolic detoxification of insecticides, including three carboxylesterases (COEAE5G, COEAE6G, Ace1), three gluta-thione *S*-transferases (GSTD3, GSTD12, GSTMS2) and 17 P450s (CYP4H17, CYP6AA1, CYP6AA2, CYP6AG1, CYP6AG2, CYP6AK1, CYP6M2, CYP6N1, CYP6P1, CYP6P2, CYP6P3, CYP6P4, CYP6P5, CYP6Z1, CYP6Z4, CYP9K1 and CYP9L1) (Fig 4A–4C). Other genes putatively implicated in insecticide resistance and overexpressed in all three field sites were eight UGTs, one glutathione peroxidase (GPXH3) and six cuticular proteins (Figs 3A–3C and S3B). Additionally, six $H^+$-transporting ATP synthases (ATPases), two trypsins, the tran-scription factor *Maf-S* and 18 other genes for which we found annotated orthologues in the *An*. *gambiae* s.l. complex were commonly up-regulated (Figs 3A–3C and S3B and S4 Table). According to the gene set enrichment analysis (GSEA) combined with a Fisher's exact test, P450s, $H^+$-transporting ATPases and UGTs were significantly enriched in the Agboville popu-lation (S2B Table). In the Dabou population, P450s, $H^+$-transporting ATPases and cuticular proteins of the CPLC family were enriched, while only CPLC cuticular proteins were enriched in the Tiassalé population.

## Comparing gene expression levels between RNA-seq and RT-qPCR

We validated the fold changes in gene expression obtained through differential gene expres-sion analysis of the RNA-seq data by subjecting RNA of the same individuals to RT-qPCR, tar-geting eight different metabolic resistance loci. As shown in Fig 5, we found a tight linear relationship between the two methods ($R^2$ = 0.923, p < 0.001), indicating good agreement. Especially for the samples from Dabou and Tiassalé, RT-qPCR tended to measure higher fold changes than RNA sequencing. Comparing each field population against both laboratory colo-nies, we observed the strongest overexpression for CYP6P3, ranging from 23-fold up with both methods for Agboville to 48-fold up with RT-qPCR for Tiassalé, while RNA-seq esti-mated a 28-fold up regulation for the same Tiassalé samples. Neither RNA-seq nor RT-qPCR estimated significant differences in CYP4G16 and GSTE2 expression levels between field and laboratory samples.

## Target-site mutations

In addition to investigating differential gene expression, we were also testing for mutations at known insecticide target sites (Fig 6 and S6 Table). We estimated the mutant allele frequency for three *kdr*-loci and the acetylcholine esterase *Ace1* locus by genotyping DNA from the legs of 40 insecticide-unexposed individuals per field population using qPCR. We detected the PY and DDT resistance-associated *kdr* allele 995F at a frequency of 52%, 75% and 60% in Agbo-ville, Dabou and Tiassalé, respectively (Fig 6A). Further, we found the CB and OP resistance-associated *Ace1* allele 280S at a frequency of 54%, 54% and 44% in Agboville, Dabou and Tias-salé, respectively (Fig 6B). We observed more heterozygous mosquitoes for the *Ace1* than for the *kdr* mutation. Notably, all tested mosquitoes from Dabou carried at least one mutant 280S allele. In Fig 6C, we report if and in which combination these alleles occurred in all 55 RNA-

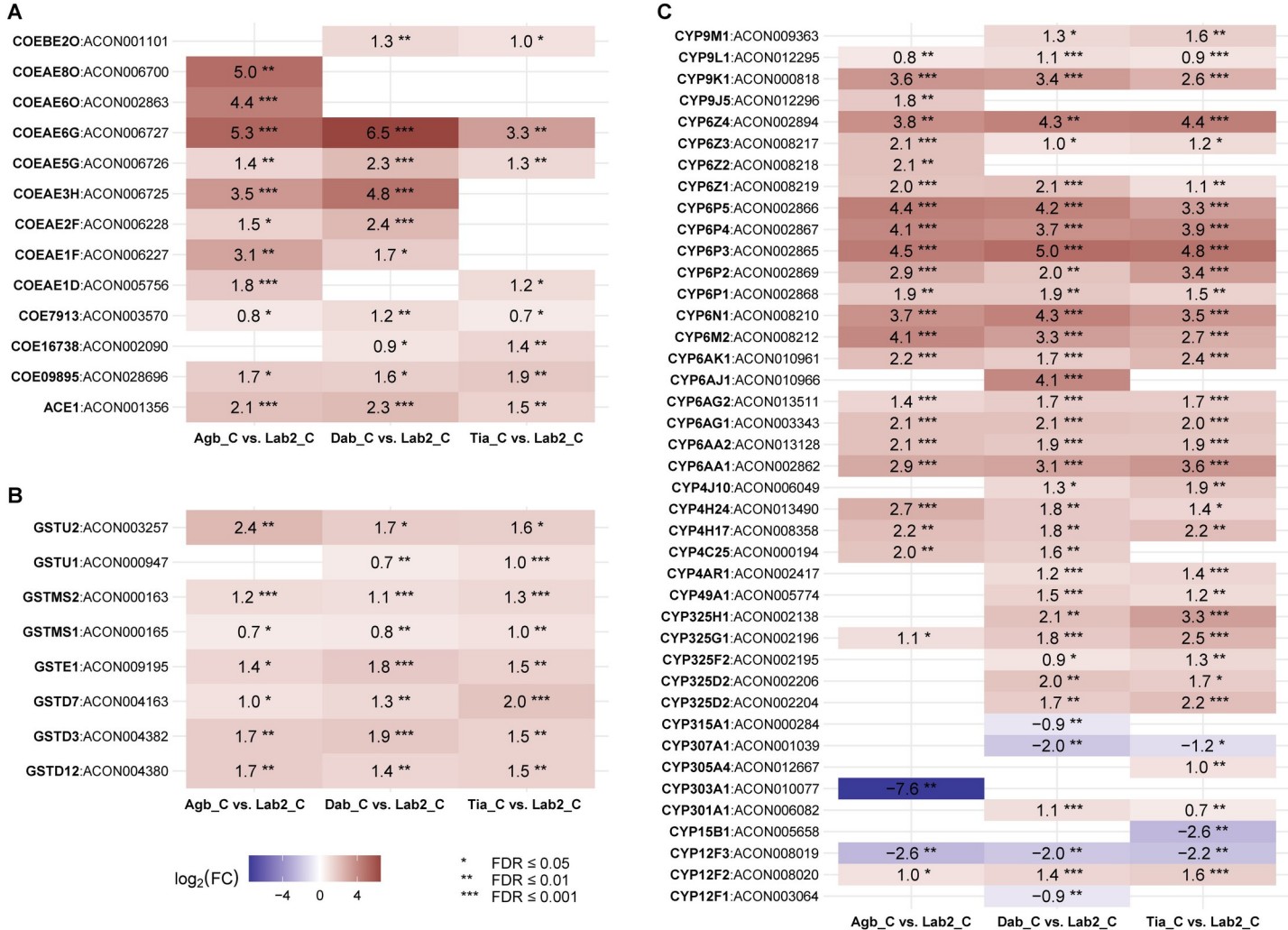

**Fig 4. Heat maps visualising differential expression of the three major insecticide detoxifying gene families between multi-resistant Ivorian (Agb_C, Dab_C, Tia_C) and susceptible laboratory colonies (Lab2_C). (A)** *Ace1* and 12 out of 41 carboxylesterases (COEs); **(B)** 8 out of 33 glutathione *S*-transferases (GSTs); and **(C)** 41 out of 110 cytochrome P450-dependent monooxygenases (P450s) for which **FDR ≤ 0.01 in at least one comparison. The number displayed on the coloured tiles shows the log₂ fold change (FC) with tiles in red depicting overexpression (log₂FC > 0) and blue underexpression (log₂FC < 0). Levels of significance *FDR ≤ 0.05; **FDR ≤ 0.01; and ***FDR ≤ 0.001. Tiles were left empty when FDR > 0.05.

sequenced individuals. The *kdr*-995F mutant allele frequency among the RNA-sequenced unexposed controls was 70% for Agboville and Dabou, whereas it was 50% for Tiassalé. The same RNA-sequenced specimens were all heterozygous for the *Ace1* mutation (50%). Notably, all tested field specimens carried neither the mutant *kdr* 995S nor the 1570Y allele. Moreover, all four mutations were absent in the tested specimens from both the susceptible Mali-NIH and Ngousso laboratory colonies (Fig 6C).

In addition to genotyping by qPCR, we performed variant calling on the RNA-seq reads with the aim to reveal single nucleotide polymorphisms (SNPs). For the majority of the samples, read coverage of the *Vgsc* gene was not high enough to confidently identify SNPs. However, for 41 out of 55 samples the coverage at the diagnostic *Ace1* locus was sufficiently high to verify if the individuals carried the non-synonymous SNP (G̲GC > A̲GC) that leads to the glycine to serine substitution. For 36 samples, the variant calling results agreed with qPCR results. For five samples, we interpreted the qPCR result as heterozygous (A/G), while the variant

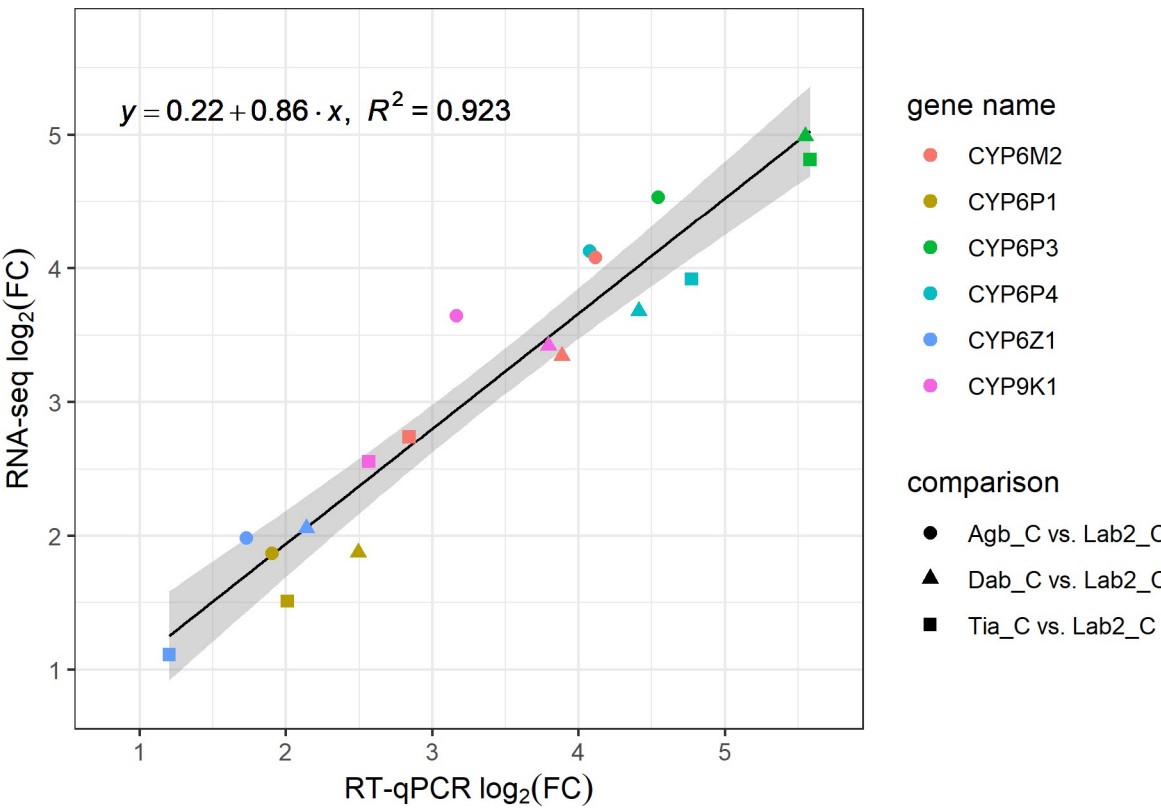

**Fig 5. Linear relationship in fold changes measured in P450s between RNA-seq and RT-qPCR.** The RT-qPCR $\log_2$ fold change ($\log_2$FC) is predictive of the RNA-seq $\log_2$FC ($R^2 = 0.923$, p < 0.001). Comparison groups were insecticide-unexposed mosquitoes from the field (n = 5 for each site: Agb_C, Dab_C and Tia_C) against all unexposed mosquitoes from the two laboratory colonies Mali-NIH and Ngousso as a group (n = 10, Lab2_C).

calling resulted in the homozygous mutant genotype (A/A). Interestingly, the qPCR quantification cycle (Cq) values of the mutant (A)-specific probe were lower than the Cq value of the (G)-specific, wild-type probe. This suggests that these individuals carried more copies of the mutant than the wild-type allele and that the latter was not detected by variant calling. Additionally, this probably means that we are underestimating the actual mutant *Ace1* allele frequencies by calculating them from the mere homo- and heterozygosity qPCR outcomes (Fig 6B).

## Discussion

The three Ivorian *An. coluzzii* field populations were all resistant to bendiocarb (CB), deltamethrin (PY) and DDT (OC), but not, or only marginally, to malathion (OP). By analysing the transcriptome of individual *An. coluzzii* females, we measured marked differences in gene expression between laboratory-reared, multi-resistant field populations and susceptible colonies, with a vast number of affected genes belonging to protein families known to be involved in insecticide resistance. Additionally, point mutations at two insecticide target sites (*Vgsc*-L995F and *Ace1*-G280S) were present in the field populations, further contributing to multiple insecticide resistance.

Resistance to multiple insecticides from different classes has previously been reported from several sites in Côte d'Ivoire [30,31], particularly from Tiassalé in 2011 [33], from Agboville in 2013 [31] and from Dabou in 2015 [71]. The extreme deltamethrin resistance intensity ($RR_{50}$

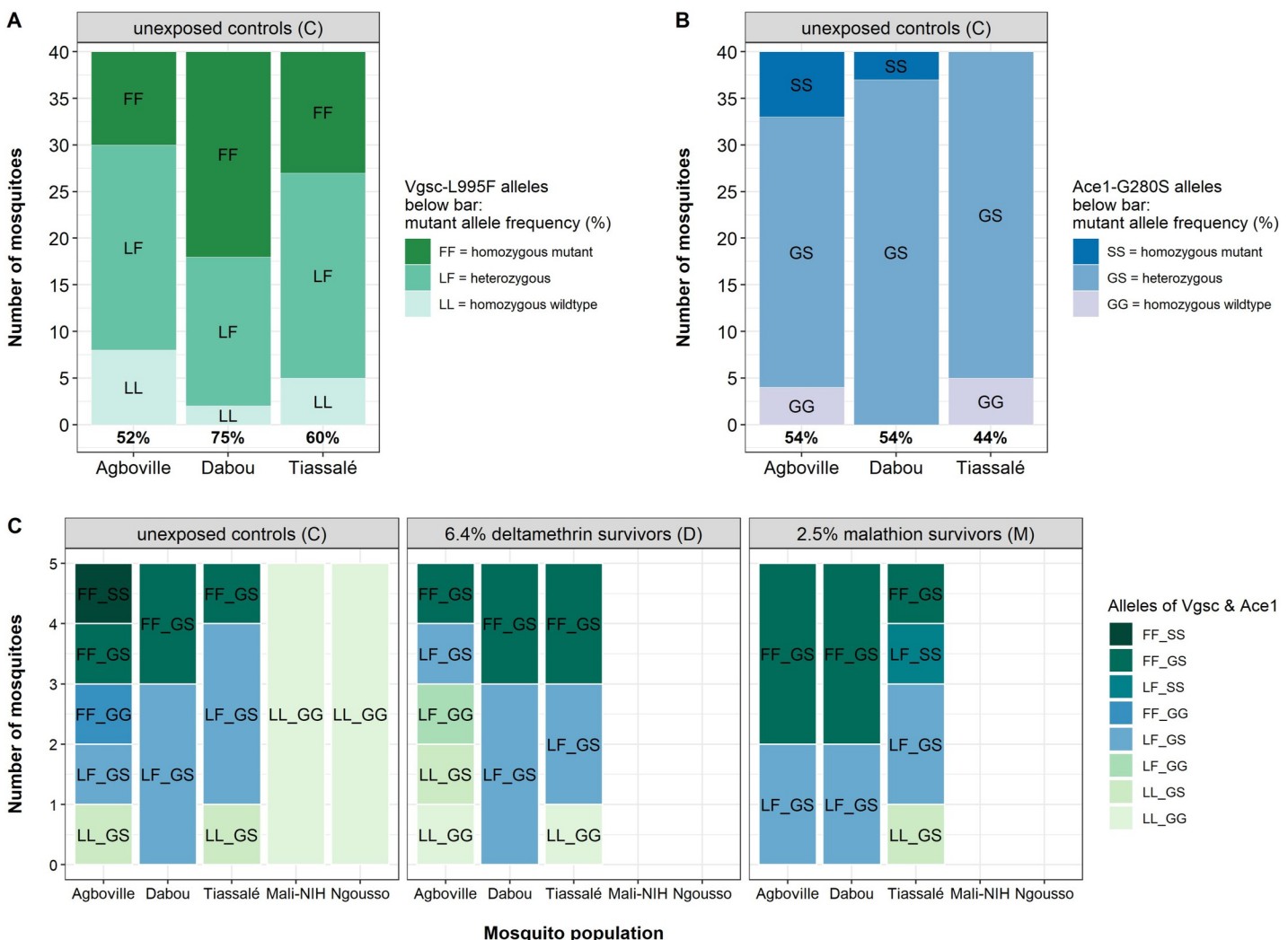

**Fig 6. Bar plots showing qPCR results on leg DNA for two target-site mutations. (A)** *Vgsc*-L995F mutation and **(B)** *Ace1*-G280S mutation frequency (% below bar) in 40 insecticide-unexposed individuals per field population. Panel **(C)** depicts in which combination the resistance-associated alleles occurred in the 55 RNA-sequenced individuals. See S4 Fig for the combined *Vgsc*-L995F and *Ace1*-G280S genotype of the 40 individuals per field population.

= 389 to 1,803) we observed in southern Côte d'Ivoire is among the highest reported to date. Similar deltamethrin resistance levels have been reported from Burkina Faso ($RR_{50}$ = 1,847, predominantly in *An. coluzzii*) [27] and from central Côte d'Ivoire ($RR_{50}$ = 1,441 to 2,405, predominantly in *An. gambiae* s.s.) [28].

Differential gene expression analysis revealed that numerous P450s were overexpressed in all three Ivorian field populations compared to the susceptible laboratory colonies (Fig 4C). An *in vitro* study using heterologously expressed P450s has shown that five of these – CYP6M2, CYP6P2, CYP6P3, CYP6P4 and CYP6P5 – can deplete PYs including deltamethrin and OPs such as malathion [25]. However, it has not yet been elucidated at what ratios the P450s-mediated malathion metabolism results in the active, highly toxic malaoxon form or in the inactive oxidative cleavage product. A recent *in vivo* study found that CYP6M2 upregulation in a susceptible background rendered transgenic *An. gambiae* resistant to PYs while they became more susceptible to malathion than the respective controls, proposing CYP6M2 to induce negative cross-resistance [36]. The results of these *in vitro* and *in vivo* studies suggest

that the same P450s can detoxify PYs and activate OPs. Intriguingly, by quantifying resistance levels using dose-response bioassays, the Tiassalé population exhibited the highest deltamethrin resistance levels, while it was at the same time the most susceptible population to malathion. This indicates that we might be observing negative cross-resistance caused by the overexpression of PY-detoxifying cytochrome P450s that are also capable of activating malathion, thereby increasing OP susceptibility. Assogba et al. demonstrated that homozygosity of the *Ace1*-G280S mutation alone is enough to evoke OP (including malathion) and CB cross-resistance in adult *An. gambiae* s.s. with an otherwise susceptible genetic background [72]. The negative cross-resistance hypothesis is thus further strengthened by the fact that we did observe only weak – if at all – resistance to malathion, despite high frequencies of the *Ace1* mutation. Additionally, several COEs which have previously been associated with OP resistance [73] were overexpressed in the field populations; yet these *An. coluzzii* were sensitive to malathion.

Based on the dose-response curves and resulting RR$_{50}$s, we would have expected to see the greatest P450 upregulation associated with negative cross-resistance in Tiassalé. However, we found no P450s, nor other genes known to be involved in insecticide resistance, differentially expressed in any comparison among the Ivorian *An. coluzzii* populations. Further, only few other P450s were overexpressed at higher levels in Tiassalé than in Agboville and Dabou when comparing each population to the laboratory colonies (Fig 4C). The absence of significant differences in gene expression between and within field population may be due to constitutive overexpression of metabolic resistance genes. In addition, the observed phenotypic response of the mosquitoes to malathion, e.g. the here observed susceptibility in Tiassalé or weak resistance in Dabou, constitutes a balance of detoxification and activation by the same enzyme(s) as indicated in the previous paragraph.

We speculate that negative cross-resistance might be compensated for by additional resistance mechanisms which decrease malathion susceptibility despite the overexpression of negative cross-resistance mediating P450s. Our transcriptomic analysis revealed that the gene encoding the target site of CBs and OPs, the acetylcholinesterase *Ace1*, was expressed higher in Dabou (FC = 4.8) and Agboville (FC = 4.4) than in Tiassalé (FC = 2.9) (Fig 4A). Additionally, we detected the mutant *Ace1* allele 280S at higher frequencies in Dabou and Agboville than in Tiassalé (Fig 6B). In summary, both results hint at higher copy numbers of the CB- and OP-resistance associated *Ace1* mutation in Dabou and Agboville and are consistent with the phenotypic bendiocarb resistance and the lower malathion susceptibility observed in these two sites.

In addition, the FCs of several overexpressed COEs were higher in Agboville and Dabou than in Tiassalé, particularly for COEAE6G and COEAE5G (Fig 4A). Carboxylesterases may confer resistance to OPs by sequestering oxon analogues and thus decreasing the amount of the highly toxic metabolite at the acetylcholinesterase target site [73]. Upregulated transcription, alone or in combination with gene amplification, has been reported to cause increased levels of esterases in resistant mosquitoes [73,74]. Another interesting candidate is COEAE3H, as it was highly overexpressed in the two populations that were less sensitive to malathion with a FC of 11.3 for Agboville and 28.2 for Dabou, while it was not differentially expressed in the Tiassalé specimens. All three COEs mentioned in this paragraph have been previously reported to be overexpressed in insecticide resistant *An. coluzzii* field populations from West Africa [26,32,75]. Functional validation of these COEs will aid elucidating whether they are indeed able to bind malathion, bendiocarb and other insecticides with high affinity, thereby inducing sequestration-mediated resistance.

We found eight *An. coluzzii* GSTs significantly overexpressed in at least one Ivorian field population compared to the susceptible colonies (Fig 4B). Although the well-characterised

DDT-metabolising GSTE2 [76,77] was not among them, other GSTs might likewise catalyse the dehydrochlorination of DDT [78,79] or be otherwise involved in the detoxification of different insecticides. Interestingly, GSTs may confer resistance to OPs as recently demonstrated for three GSTs of the oriental fruit fly *Bactrocera dorsalis* that can deplete malathion and its highly toxic metabolite malaoxon [80,81]. Whether such a GST-mediated OP detoxification mechanism also exists in malaria vectors remains to be elucidated.

In addition to the mentioned P450s, COEs and GSTs, we found genes overexpressed belonging to other protein families putatively involved in insecticide resistance, including the transcription factor *Maf-S*, several UGTs, H$^+$-transporting ATPases and cuticular proteins. However, we did not find any CSPs commonly upregulated in the multi-resistant field populations (S3F Fig) which recently have been elucidated as a novel resistance mechanism sequestering pyrethroids in mosquitoes colonised from Tiassalé [19]. Interestingly, in our study the transcription factor *Maf-S* was 1.6, 1.9 and 2.2 times higher expressed in Agboville, Dabou and Tiassalé specimens, respectively, than in the susceptible controls. The transcription factor *Maf-S* has been shown to control the expression of key insecticide detoxification genes, including *CYP6M2*, and reducing *Maf-S* transcript levels through RNAi-mediated knockdown significantly increased deltamethrin mortality while decreasing malathion mortality [65]. Here, the observed upregulation of *Maf-S* that has been shown to reverse the phenotype between pyrethroids and the pro-insecticide malathion strengthens the evidence for negative cross-resistance in our Ivorian field populations.

Uridine diphosphate (UDP)-glycosyltransferases (UGTs) are enzymes that catalyse glucosidation by the addition of glucose to lipophilic chemicals, making these toxins more water-soluble and thus facilitating their excretion [20]. Our RNA-seq analysis revealed that eight out of 17 UGTs were significantly overexpressed in all three multi-resistant field populations compared to the susceptible colonies (Figs 3A–3C and S3B and S3C). Zhou and colleagues proposed UGT308 and UGT302 families to be main candidates associated with pyrethroid resistance in *An. sinensis* from China [82]. Interestingly, the most up-regulated UGTs in our study also belong to these two families and might likewise contribute to the phenotypic PY resistance.

In addition, numerous H$^+$-transporting ATPase transcripts were consistently 1.4 to 3 times more abundant in the field populations than in the lab colonies (Figs 3A–3C and S3D), and according to the GSEA they were significantly enriched in Agboville and Dabou. A recent meta-analysis on microarray data of African malaria vectors found the subunit epsilon of the F-type ATPase (ACON006879) overexpressed across all *An. coluzzii* studies and functionally validated its possible involvement in pyrethroid resistance [83].

Finally, numerous transcripts encoding for cuticular proteins were highly enriched in the field populations, most dominantly in Dabou and Tiassalé (Figs 3A–3C and S3E). Should the observed higher cuticular protein expression result in a thicker or otherwise altered cuticle, insecticide uptake may be reduced. Slower insecticide uptake would allow more time for metabolic enzymes to break down insecticides and thus act favourably both on i) multiple resistance mediated by detoxification or sequestration; and ii) on negative cross-resistance resulting in the inverse increased toxic effect through the enzymatic activation of pro-insecticides. The cuticular protein CPLCP25 was among the top 20 overexpressed genes in all three field sites and merits further investigation.

The small sample size and high variability in gene expression levels between specimens is a limitation of our RNA-seq experiment in that small differences cannot be called at high confidence. However, despite the small sample size, we could still confirm the overexpression of several P450s such as CYP6M2 [24], CYP6P3 [23] and CYP9K1 [84] that have been validated as markers for metabolic resistance and have previously been found significantly

overexpressed in microarrays on pooled Ivorian *Anopheles* specimens [28,32]. In addition, the good agreement between RT-qPCR and RNA-seq data on six P450 detoxification markers demonstrates the validity of the approach for differential gene expression quantification. The advantage of transcriptome sequencing of individual mosquitoes rather than pools is that results from pools could be misleading if variation among individuals is inconsistent yet buried in a single value. Importantly, sequencing individuals also allows investigating sequence variations such as SNPs, insertions or deletions in the protein-coding region. However, due to the limited amount of RNA and lower sequencing depth, these gains come at the cost of weaker statistical power in detecting small differences in gene expression.

Malaria vectors resistant to multiple insecticide classes pose a big challenge for mosquito control as they leave very little scope for IRM strategies in the anyhow small insecticide toolbox. This is the first time that indications of P450-mediated negative cross-resistance are observed in *Anopheles* field populations. Negative cross-resistance could be exploited to benefit vector control, as the targeted use of pro-insecticides will have a more toxic effect against multi-resistant mosquitoes if they express the right P450s at higher levels compared to their more susceptible counterparts. A strategy could be to use OP-based IRS, such as pirimiphosmethyl or malathion, against PY-resistant malaria vectors alone or in combination with PY-treated bed nets.

In conclusion, transcriptome and target-site mutation analysis revealed the simultaneous presence of a wide range of known and putatively new insecticide resistance mechanism in multi-resistant *An. coluzzii* field populations from southern Côte d'Ivoire. Despite the presence of target-site mutations, the field populations were remarkably susceptible to malathion. A potential explanation for the observed phenomenon is negative cross-resistance mediated by overexpressed P450s that may activate pro-insecticides, thereby increasing malathion susceptibility.

## Supporting information

**S1 Fig. Experimental design for RNA sequencing.** We sequenced five individual mosquitoes (i.e. five biological replicates) per *An. coluzzii* population and experimental condition. The five laboratory-reared *An. coluzzii* populations were three multi-resistant populations from southern Côte d'Ivoire collected at the larval stage in Agboville (Agb), Dabou (Dab) and Tiassalé (Tia) and two insecticide susceptible laboratory colonies Ngousso and Mali-NIH (Lab2). The three experimental conditions were insecticide-unexposed control (C), selected against 6.4% deltamethrin (D) and 2.5% malathion (M). One biological sample denotes RNA extracted from an individual, laboratory-reared, 3- to 6-day-old female *An. coluzzii* mosquito. Each mosquito was sequenced twice in different runs, producing two technical replicates. NA, not available. (PPTX)

**S2 Fig. Multi-dimensional scaling (MDS) plots showing the relative similarities between expression profiles of different samples. (A)** MDS plot showing lower variation between technical replicates (yellow: run1 and blue: run2) than between biological replicates (labels removed for better visibility). **(B)** MDS plot using summed read counts of the two runs showing clear separation between two lab colonies and field populations (Agb, Dab, Tia) and also between the Mali-NIH and Ngousso lab colonies, but not between the three field populations. **(C)** MDS plot showing that there were no obvious patterns indicative of experimental condition: insecticide-unexposed control (C), selected against 6.4% deltamethrin (D), and 2.5% malathion (M). (PPTX)

**S3 Fig. Additional heat maps visualising differential expression between multi-resistant Ivorian specimens (Agb, Dab, Tia) and the susceptible laboratory colonies (Lab2). (A)** Commonly underexpressed genes; **(B)** commonly overexpressed genes apart from the three major detoxification enzyme families shown in Fig 4; **(C)** Uridine diphosphate UDP-glycosyl-transferases (UGTs); **(D)** $H^+$-transporting ATP synthases; **(E)** cuticular proteins; **(F)** chemo-sensory proteins (CSPs); **(G)** ATP-binding cassette (ABC) transporters; and **(H)** 41 P450s, in addition to the unexposed controls also including the insecticide selected samples compared to the laboratory colonies. On the left of the heat map are the putative, orthology-derived gene names and ACON gene IDs for which $^{**}FDR \leq 0.01$ in at least one comparison (except for CSPs). The number displayed on the coloured tiles shows the $log_2$ fold change ($log_2FC$) with tiles in red depicting overexpression ($log_2FC > 0$) and blue underexpression ($log_2FC < 0$). Levels of significance $^*FDR \leq 0.05$; $^{**}FDR \leq 0.01$; and $^{***}FDR \leq 0.001$. Tiles were left empty when $FDR > 0.05$, except for S3F Fig for CSPs includes $FDR > 0.05$, not significant (ns). (PPTX)

**S4 Fig. Bar plot showing combined *Vgsc*-L995F and *Ace1*-G280S genotypes of Ivorian field populations.** Bar plots showing the combined qPCR results on leg DNA for two target-site mutations (*Vgsc*-L995F and *Ace1*-G280S), i.e. in which combination the resistance-associated alleles occurred in 40 insecticide-unexposed individuals per field population. (PPTX)

**S1 Table. Supplementary information on (RT-)qPCR assays. (A)** Primer and TaqMan probe sequences with their 5' and 3' modifications and concentrations used for the (RT-)qPCR assays in this study. **(B)** List of abbreviations, suppliers, thermal cycling parameters and references for the primers and probes listed in S1A Table. (XLSX)

**S2 Table. Annotation list *Anopheles coluzzii* Ngousso (ACON) genes and the results of the gene set enrichment analysis (GSEA). (A)** Full annotation list of the 13,299 ACON genes based on orthology and blastp search as well as the GSEA. **(B)** Detailed result of the GSEA. We regarded a gene set as significantly enriched only when the FDR of both the GSEA and Fisher's exact test was $\leq 0.01$. (XLSX)

**S3 Table. Differential gene expression analysis results for relevant comparisons between or within the Ivorian field populations including annotations, $log_2$ fold changes and FDR for 10,519 out of the 13,299 ACON genes.** (XLSX)

**S4 Table. Differential gene expression analysis results for all comparisons between each of the three multi-insecticide resistant field populations and the two susceptible laboratory colonies as a group including annotations, $log_2$ fold changes and FDR for 10,519 out of the 13,299 ACON genes.** (XLSX)

**S5 Table. Expression levels of eight metabolic resistance loci in the three multi-resistant field populations from Côte d'Ivoire relative to two susceptible laboratory colonies measured with RT-qPCR.** The housekeeping gene encoding the ribosomal protein S7 (RPS7) internal calibrator in each triplex reaction. Expression levels were analysed using the REST 2009 software v2.0.13. (XLSX)

**S6 Table. Genotype of each individual mosquito for all analysed insecticide resistance loci.** Highlighted in red is the individual (ID: 01_Agb_D02) that we excluded from the RNA-seq analysis because we obtained a low number of reads and low mapping rates in both sequencing runs.
(XLSX)

**S7 Table. Insecticide bioassay data sets and result tables. (A)** Bioassay data using WHO discriminating concentrations to measure phenotypic insecticide resistance in *Anopheles coluzzii* from southern Côte d'Ivoire. Note that for the Tiassalé population we observed 100% mortality 24 h post malathion exposure, but in order to estimate a 95% CI with GLMs, we artificially added one survivor highlighted in red in this model input data set. **(B)** Outcome of the standard WHO insecticide susceptibility tests using discriminating concentrations of four insecticides. Numeric data of average mortalities and their 95% confidence intervals (CIs) as estimated with GLMs and plotted in Fig 2A. **(C)** Data of dose-response bioassay to measure the degree of phenotypic insecticide resistance against deltamethrin and malathion in *Anopheles coluzzii* from southern Côte d'Ivoire.
(XLSX)

**S1 Data. Raw read count dataset.** Raw read counts for each of the 13,299 ACON genes and technical replicate of the 55 biological mosquito samples (n = 110).
(TXT)

**S2 Data. Metadata for raw read count dataset.** Metadata for each RNA-sequenced sample including information on sequencing run, mosquito population, experimental condition (insecticide or control), environment (field or lab population), group (five biological replicates for each experimental condition and population).
(TXT)

## Acknowledgments

We would like to thank the following people for their inputs and fruitful discussion for the RNA-seq experimental design: Vasileia Balabanidou and Panagiotis Ioannidis from FORTH-IMBB, Monica Golumbeanu and Natalie Wiedemar from Swiss TPH and Geoffrey Fucile from sciCORE. Thanks to Mark Hoppé from Syngenta Crop Protection, Basel, Switzerland we could still use silicon oil for preparing insecticide treated filter papers when worldwide distribution of the oil was interrupted. We acknowledge the farmers from Agboville, Dabou and Tiassalé for their permission to collect mosquito larvae in their crop fields as well as Sébastien Oyou Kere and Williams Adienin for helping with the larval collections. We are grateful to Dobri Laurent Didier, Akoupo Alain Sylvain and late Nestor Kesse for their technical assistance at the CSRS insectary. We are thankful to Sara Mitchell from Verily Life Sciences (Google Inc.) for kindly sharing her laboratory protocols for nucleic acid extraction from mosquitoes. We thank George Papagiannakis and Pantelis Topalis from FORTH-IMBB for their bioinformatics support and collaboration through Infravec2. Parts of the calculations were performed at sciCORE (http://scicore.unibas.ch/) scientific computing core facility at University of Basel, with support by the SIB—Swiss Institute of Bioinformatics. We gratefully acknowledge Silvan Hälg from Swiss TPH for critically reading earlier versions of the manuscript and providing valuable input. The following reagent was obtained through BEI Resources, NIAID, NIH: *Anopheles coluzzii*, Strain Mali-NIH, Eggs, MRA-860, contributed by Nora J. Besansky.

## Author Contributions

**Conceptualization:** Nadja C. Wipf, Chouaïbou S. Mouhamadou, John Vontas, Pie Müller.

**Data curation:** Nadja C. Wipf, Wandrille Duchemin.

**Formal analysis:** Nadja C. Wipf, Wandrille Duchemin, Pascal Mäser.

**Funding acquisition:** Nadja C. Wipf, Konstantinos Mavridis, John Vontas, Pie Müller.

**Investigation:** Nadja C. Wipf, France-Paraudie A. Kouadio, Behi K. Fodjo, Christabelle G. Sadia, Laura Vavassori.

**Methodology:** Nadja C. Wipf, Wandrille Duchemin, Pascal Mäser, Pie Müller.

**Project administration:** Nadja C. Wipf, Pie Müller.

**Resources:** Nadja C. Wipf, Wandrille Duchemin, Konstantinos Mavridis.

**Software:** Wandrille Duchemin.

**Supervision:** Pie Müller.

**Validation:** Nadja C. Wipf, Konstantinos Mavridis.

**Writing – original draft:** Nadja C. Wipf.

**Writing – review & editing:** Nadja C. Wipf, Wandrille Duchemin, France-Paraudie A. Kouadio, Behi K. Fodjo, Christabelle G. Sadia, Chouaïbou S. Mouhamadou, Laura Vavassori, Pascal Mäser, Konstantinos Mavridis, John Vontas, Pie Müller.

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
