## [Decision Letter · Decision Letter 0]

29 Apr 2021

Dear Dr Wipf,

Thank you very much for submitting your Research Article entitled 'Multi-insecticide resistant malaria vectors in the field remain susceptible to malathion, despite the presence of Ace1 point mutations' to PLOS Genetics.

The manuscript was fully evaluated at the editorial level and by independent peer reviewers. The reviewers appreciated the attention to an important problem, but raised some substantial concerns about the current manuscript. Based on the reviews, we will not be able to accept this version of the manuscript, but we would be willing to review a much-revised version. We cannot, of course, promise publication at that time.

If you decide to revise the manuscript for further consideration at PLOS Genetics, please aim to resubmit within the next 60 days, unless it will take extra time to address the concerns of the reviewers, in which case we would appreciate an expected resubmission date by email to plosgenetics@plos.org.

[LINK]

We are sorry that we cannot be more positive about your manuscript at this stage. Please do not hesitate to contact us if you have any concerns or questions.

Yours sincerely,

Subba Reddy Palli, Ph.D.

Associate Editor

PLOS Genetics

Scott Williams

Section Editor: Natural Variation

PLOS Genetics

Reviewer's Responses to Questions

**Comments to the Authors:**

Reviewer #1: Really interesting and informative paper on resistance mechanisms in three field caught An. coluzzii populations exposed to both deltamethrin and malathion. The authors have done an excellent and thorough analysis and I enjoyed reading the paper. I have some comments below.

Major comments:

All R code should be available on a repository to ensure repeatability

Why was this done with Acol N1.0 and not PEST, when PEST is the better annotated and curated genome/transcriptome? This seems to add so much unneccessary complexity to this analysis. In my opinion, to ensure robustness of the annotations and to keep this consistent with literature, it should have been done against the PEST genome.

I am confused about S3 - I think different tables are needed for each of the comparisons - insecticide exposed vs control, then controls vs each other and Delta exposed vs Mal exposed for each population - they are asking such different questions and lumping them into one table is hard to follow.

In a similar vein to above, the S4 table would be much more accessible if it was split into different populations, or if the p-value was given for each in the columns rather than having multiple rows.

S4 needs to be split into control vs susceptibles and exposed vs susceptibles regardless of the above.

It would be good to have a supplementary table with the genotypes for the mutations for each individual

Minor comments:

Line 27: enables optimising vector - enambles optimisation of vector

Line 53/54: I think it is a reach to say that this is the first time p450-mediated negative cross resistance is observed in field - you aren't sure it is p450s causing this susceptibility. It is likely but not shown.

Line 75: Should mention CSP mediated resistance to highlight it is not just p450s mediating this - pretty important for the conclusions esp. as you mention UGTs/ABCs which are more tenuously linked

Line 82: Should mention I1527T as an alternative to 995F

Line 200/201: How were these concentrations chosen?

208: You sequenced individual females? This isn't clear

237: Were 1-3 legs removed from the susceptible populations too? It isn't clear whether the susceptible populations were checked for species/target site

255: It is unclear here whether you combined those 55 individuals or did a run for each until reading the results.

305: How many mosquitoes were in one biological rep and how many biological reps were done? Did you use RNA directly for qPCR or cDNA?

More of an interest question, did the authors test against any other OP class?

Reviewer #2: The article titled "Multi-insecticide resistant malaria vectors in the field remain susceptible to malathion, despite the presence of Ace1 point mutations" is an interesting piece of work from experts in the field of insecticide resistance. Negative cross-resistance to OPs insecticide (malathion) in malaria vector populations even though ACE1 mutation significantly presents at high frequency in these populations is interesting. This work benefited from the addition of RNA sequencing data that provides good genetics resources and link genotyping with phenotyping data. This also provide more insight into insecticide resistance mechanism by defining the genes and pathways that are more likely to drive a high level of resistance to deltamethrin, DDT and bendiocarb. I agree with the authors that Cytochrome P450s are more likely the core enzymes that operate in this populations and detoxify deltamethrin and Bendiocarb. Also, I agree that these strains remain susceptible to malathion questioned cross-resistance within OPs. I hoped to see that the authors considered other OPs as a positive control to justify their finding in this context. In other words, is the observed negative cross-resistance is reported/examined against other OPs that are operationally recommended for fighting malaria in Africa, such as Primiphos-methyl? This is quite important as it should give operational and structural insight into the effect of Ace1-G280S on the cross-resistance mechanism within OPs.

The authors attributed the observed negative cross-resistance against malathion to a high level of oxidative enzymes, which increase the toxicity of the pro-insecticide. Although this is theoretically acceptable, it will be more supportive of including synergistic assay with cytochrome P450s inhibitors (e.g. PBO) to see if the inhibition of oxidative enzyme will decrease the toxicity against malathion in these populations. The ideal is to run the experiment against all insecticides and extended it to all populations in the presence and absence of PBO or similar inhibitors. Still, the minimum set of data against malathion, primiphos-methyl and or bendiocarb remain acceptable.

Reviewer #3: The manuscript describes an excellently performed and described transcriptomic analysis of An.colluzzi strains from Cote D'Ivoire that were shown in the manuscript to display negative cross resistance to pyrethroids and malathion. It has long been suspected that such resistance phenotypes are feasible due to the pro-insecticide nature of malathion that is activated to more toxic compounds by the same P450 enzymes known to detoxify pyrethroids. The manuscript clearly shows that this form of pyrethroid/malathion sensitivity is present in mosquitoes within the Ivory Coast.

The execution of the work is exemplary and I don't have any issues with how it has been performed. The conclusions drawn are valid and go as far as they can really, since the evidence is one of correlation rather than causation. Pharmacodynamic analysis of malathion metabolism would add further weight to the work, but is very difficult to perform on such insects.

There are however three areas that need to be considered that are not developed within the discussion, and need to be addressed.

The mosquitoes do show low levels of resistance to malathion, and the authors suggest that this may be due to the presence of ace1 mutations in the strains and other potential mechanisms causing decrease in susceptibility. There is considerable work done with ace1 mutations that is not discussed, such as Assogba et al 2014, that have data on resistance levels induced by ace1 mutations alone - how do these compare with the resistance shown in the these strains. what level of resistance is seen in other strains carrying ace1 mutations. do other mechanisms need to be invoked to complement ace1?

Alternatively, all 3 lines show high levels of cyp6P3 expression and it is feasible that such P450s also detoxify the oxon form of malathion, and so sensitivity of the mosquitoes to malathion will be a balance of activation and detoxification by the same enzyme. This needs addressing too.

The manuscript also describes several genes and gene families that are differentially regulated in the strains showing negative cross resistance compared to a lab strain. In the Ingham transcriptomic paper, BMC Genomics2017 doi: 10.1186/s12864-017-4086-7, they show that down regulation of the Maf-S transcriptions factor causes a reversal of a negative cross resistance phenotype. It would be good to compare the genes regulated in the current study with those in Ingham et al to identify commonalities that would strengthen the evidence provided.

**Have all data underlying the figures and results presented in the manuscript been provided?**

Reviewer #1: **No: **Need the R code

Reviewer #2: Yes

Reviewer #3: Yes

PLOS authors have the option to publish the peer review history of their article (what does this mean?). If published, this will include your full peer review and any attached files.

Reviewer #1: No

Reviewer #2: No

Reviewer #3: No

---

## [Decision Letter · Decision Letter 1]

23 Nov 2021

Dear Dr Wipf,

We are pleased to inform you that your manuscript entitled "Multi-insecticide resistant malaria vectors in the field remain susceptible to malathion, despite the presence of Ace1 point mutations" has been editorially accepted for publication in PLOS Genetics. Congratulations!

Yours sincerely,

Subba Reddy Palli, Ph.D.

Associate Editor

PLOS Genetics

Scott Williams

Section Editor: Natural Variation

PLOS Genetics

Comments from the reviewers (if applicable):

Reviewer's Responses to Questions

**Comments to the Authors:**

Reviewer #1: I thank the authors for spending time and addressing each of my comments. I again congratulate them on a very interesting paper and recommend it for publication.

Reviewer #2: I agree with the author that the new bioassay data did not significantly explain the negative cross-resistance observed with the original colony. I can understand how difficult it would be to collect mosquitoes again to redo the bioassay and in the context of characterizing the mechanism of OPs resistance, the new bioassay data will not be entirely explained by genetic data obtained from the original colonies presented in the manuscript. Regardless, this work remains publishable and deliver an exciting data set that should give more insight into the mechanism of insecticide resistance in malaria vectors.

Reviewer #3: The authors have addressed all of the points made satisfactorily.

**Have all data underlying the figures and results presented in the manuscript been provided?**

Reviewer #1: Yes

Reviewer #2: Yes

Reviewer #3: Yes

PLOS authors have the option to publish the peer review history of their article (what does this mean?). If published, this will include your full peer review and any attached files.

Reviewer #1: No

Reviewer #2: No

Reviewer #3: No

**Data Deposition**

http://datadryad.org/submit?journalID=pgenetics&manu=PGENETICS-D-21-00312R1

**Press Queries**

---

## [Editor Report · Acceptance letter]

30 Dec 2021

PGENETICS-D-21-00312R1 

Multi-insecticide resistant malaria vectors in the field remain susceptible to malathion, despite the presence of Ace1 point mutations 

Dear Dr Wipf, 

We are pleased to inform you that your manuscript entitled "Multi-insecticide resistant malaria vectors in the field remain susceptible to malathion, despite the presence of Ace1 point mutations" has been formally accepted for publication in PLOS Genetics! Your manuscript is now with our production department and you will be notified of the publication date in due course.

With kind regards,

Zsofia Freund

PLOS Genetics

On behalf of:
